# Not Just a Banana: The Extent of Fruit Cross-Reactivity and Reaction Severity in Adults with Banana Allergy

**DOI:** 10.3390/foods12132456

**Published:** 2023-06-23

**Authors:** Narachai Julanon, Ben Thiravetyan, Chanita Unhapipatpong, Nutchapon Xanthavanij, Thanachit Krikeerati, Torpong Thongngarm, Chamard Wongsa, Wisuwat Songnuan, Phornnop Naiyanetr, Mongkhon Sompornrattanaphan

**Affiliations:** 1Division of Dermatology, Department of Medicine, Srinagarind Hospital, Khon Kaen University, Khon Kaen 40002, Thailand; naraju@kku.ac.th; 2Department of Immunology, Faculty of Medicine Siriraj Hospital, Mahidol University, Bangkok 10700, Thailand; ben.thi@gmail.com (B.T.); nutchapon1995@gmail.com (N.X.); phornnop.nai@mahidol.ac.th (P.N.); 3Division of Clinical Nutrition, Department of Medicine, Khon Kaen Hospital, Khon Kaen 40000, Thailand; chanita@kkumail.com; 4Division of Allergy and Clinical Immunology, Department of Medicine, Faculty of Medicine Siriraj Hospital, Mahidol University, Bangkok 10700, Thailand; t.krikeerati@gmail.com (T.K.); torallergy@gmail.com (T.T.); chamard.won@gmail.com (C.W.); 5Faculty of Medicine Siriraj Hospital, Center of Research Excellence in Allergy and Immunology, Mahidol University, Bangkok 10700, Thailand; 6Systems Biology of Diseases Research Unit, Faculty of Science, Mahidol University, Bangkok 10400, Thailand; wisuwat.son@mahidol.edu; 7Center of Excellence on Environmental Health and Toxicology (EHT), OPS, MHESI, Bangkok 10400, Thailand; 8Department of Plant Science, Faculty of Science, Mahidol University, Bangkok 10400, Thailand

**Keywords:** banana allergy, cross-reactivity, plant food allergies, food safety, anaphylaxis, fruit hypersensitivity

## Abstract

This cross-sectional study aimed to investigate the prevalence and clinical characteristics of cross-reactivity and co-allergy to other plant foods among adult patients with IgE-mediated banana allergy in Thailand. A structured questionnaire was used to assess clinical reactivity, and cross-reactivity diagnoses were based on reactions occurring within 2 years of banana allergy onset, within 3 h of intake, and confirmed by allergists. Among the 133 participants, the most commonly associated plant foods with clinical reactions were kiwi (83.5%), avocado (71.1%), persimmon (58.8%), grapes (44.0%), and durian (43.6%). Notably, 26.5% of the reported reactions to other plant foods were classified as severe. These findings highlight the common occurrence of cross-reactivity/co-allergy to other plant foods in banana-allergic patients, with a significant proportion experiencing severe reactions. Travelers to tropical regions should be aware of this risk and advised to avoid specific banana cultivars and plant foods with reported high cross-reactivity. The inclusion of self-injectable epinephrine in the management plan for patients with primary banana allergy should be considered due to the substantial proportion of reported severe reactions and the wide range of clinical cross-reactivity and co-allergy observed.

## 1. Introduction

Banana or *Musa* spp. belongs to the Musaceae family. It is cultivated worldwide as the fifth most popular agricultural food crop [1]. It is widely used for various purposes due to its nutritional and phytochemical properties [2]. However, banana allergy has been well established and the prevalence rate is 0.04% to 1.2% in the general population and up to 46% to 67% in patients with atopic dermatitis and asthma, respectively [3]. The spectrum of banana allergy ranges from oral allergy syndrome to anaphylaxis [4,5]. Primary sensitization to pollen is frequently the trigger for allergies to fruits and vegetables, a pollen-food allergy syndrome (PFAS) [6]. At least five proteins in bananas have been linked with cross-reactivity to multiple fruit pan-allergens, latex, ragweed, and pollens. Structural proteins (profilins), such as Mus a 1, and pathogenesis-related (PR) proteins, such as Mus a 2 (Chitin-binding proteins), Mus a 3 (lipid transfer proteins), Mus a 4 (Thaumatin-like proteins), and Mus a 5 (ß-1,3-glucanases), are probably involved in cross-reactivity between banana and other fruits [7].

Patients with banana allergy usually experience other plant food allergies, such as kiwi, avocado, chestnut, hazelnut, walnut, melon varieties, orange, persimmon, plum, tomato, and zucchini [8,9,10,11,12,13,14]. These associations and patterns of cross-reactivities are extensively documented in the literature. However, limited data are available regarding cross-reactivity patterns among tropical and exotic fruits, and these patterns may vary between regions due to potential differences in molecular sensitization and primary sensitizer profiles. Previous studies have reported allergic reactions in patients with banana allergy to other tropical and exotic fruits, such as figs and papaya, either as part of Ficus-fruit syndrome or as a response to a single fruit [15,16].

The conventional recommendation for individuals with positive skin tests or in vitro IgE test results for plant foods is to restrict the entire group of fruits to prevent cross-reactivity [17]. However, emerging evidence suggests that such a broad restriction may not always be necessary. Prick-to-prick tests (PTP) using fresh plant food are often considered superior to extract-based skin tests due to their higher sensitivity. When combined with a convincing medical history, a positive PTP to a specific fruit can confirm a plant food allergy [18]. Given the significant cross-reactivity among plant allergens, the clinical relevance of sensitization often requires confirmation through oral food challenges (OFC). However, it is important to acknowledge that OFC carries a risk of triggering an allergic reaction, particularly when the initial reaction was severe. Therefore, caution should be exercised when considering OFC in such cases. There is a lack of substantial data on PTP as a reliable predictor of clinical cross-reactivity to other plant foods in individuals with established banana allergy.

Therefore, we aimed to investigate the prevalence of self-reported clinical cross-reactivity/co-allergy of banana and other plant foods in Thai patients based on a convincing history by using a structured questionnaire. We also analyzed the correlations between mean wheal diameters for other related fruits, aeroallergens, and other plant foods.

## 2. Materials and Methods

### 2.1. Study Design

This cross-sectional study included 133 adults from the banana-allergic adult cohort (BAAC), who were prospectively recruited from Siriraj Hospital, the largest tertiary hospital in Thailand, between May 2019 and September 2022. The patient flowchart is shown in Figure 1. The study was conducted in accordance with the Declaration of Helsinki and was approved by the institutional review boards/ethics committees of Mahidol University, Thailand (approval code 127/2562(EC4)), and Khon Kaen University, Thailand (approval code HE651494). Written informed consent was obtained from all participants.

### 2.2. Participants

The study included adult patients aged 18 years or older with a relevant history of IgE-mediated banana allergy, such as oral allergy syndrome, urticaria, angioedema, and anaphylaxis within 3 h of consumption, and laboratory confirmation from either PTP or specific IgE (sIgE) to banana (>0.35 kU/L). Patients with dementia, an inability to communicate, pregnancy, lactation, immunocompromised patients, patients taking immunosuppressive drugs, and patients with autoimmune disease, celiac disease, active eosinophilic gastrointestinal disease within the last 2 years, or psychiatric disorders, were excluded from the study.

### 2.3. Laboratory Testing

Skin prick tests (SPT) using commercial extracts and PTP were performed according to standard procedures [19]. PTP was carried out with several cultivars of raw banana, including Pisang Awak, Cavendish, Leb-mue-nang, Pisang Mas, and Silver Bluggoe banana; other plant foods, including kiwi, avocado, green grape, red grape, persimmon, durian, wheat grain, gliadin, glutenin, rice berry, peanuts and soybean; and aeroallergens, including Bermuda grass, Johnson grass, *D. pteronyssinus*, *D. farinae*, kapok, cat, dog, mouse, American cockroach, German cockroach, *Penicillium*, *Aspergillus*, and *Alternaria,* on the volar side of forearm. A positive histamine control (ALK-Abello Pharmaceutical, Inc., Copenhagen, Denmark) and negative control (0.9% normal saline solution) were applied. A positive SPT was defined as a mean wheal diameter ≥3 mm at 15 min after pricking. One investigator blinded to other findings evaluated the SPT results. Serum sIgE to banana (ImmunoCAP, Phadia AB, Uppsala, Sweden) was also measured. A cut-off value > 0.35 kU/L was defined as a positive sIgE test result.

### 2.4. Data Collection and Definitions

Baseline characteristics, atopic comorbidities, characteristics of allergic reaction, reported reactions to other plant foods, types (cultivars) of consumed banana, and history of latex-related reactions were collected by interview. We collected types of specific plant foods ingested and allergic symptoms following the ingestion of each plant food after the onset of banana allergy. The specific symptoms of allergy to other plant foods were also recorded. Patient history of types of bananas associated up to the possible fourth reaction in life to banana cultivars and other fruits was collected using a structured questionnaire prospectively by interview.

Clinical cross-reactivity or co-allergy, related to primary banana allergy, was defined as the consumption of those plant foods within 2 years after the onset of banana allergy, with patients experiencing a reaction within 3 h after intake and diagnosed as cross-reactivity by an allergist. We classified the reactions into two categories: severe (anaphylaxis) and non-severe (not anaphylaxis), according to the World Allergy Organization’s (WAO) 2020 criteria. Anaphylaxis is diagnosed when there is an acute onset of skin and/or mucosal lesions accompanied by at least one of the following: respiratory compromise, hypotension, or severe gastrointestinal symptoms. If a patient experiences acute onset hypotension, bronchospasm, or laryngeal involvement upon exposure to a known allergen or highly probable allergen, the diagnosis of anaphylaxis is warranted [20].

The primary outcome was the proportions of the reported clinical reactions of banana allergy and other plant foods. Secondary outcomes were the severity of the reaction, the results of PTPs, and the correlations between Thai banana cultivars and certain plant foods along with aeroallergens.

### 2.5. Statistical Analysis

Data were analyzed descriptively. Continuous variables are presented as mean (SD) or median and interquartile range (IQR) as appropriate. Categorical variables are presented as frequency (%). Normal distribution was checked with histograms, quantile-quantile plots, and the Shapiro–Wilk test. Pairwise correlations between skin prick test mean wheal diameters were analyzed using Spearman’s rank correlation. Due to the possibility of measurement error and poor accuracy regarding demonstrating the true degree of reactivity by SPT caused by many factors (e.g., varying allergenic protein components abundance), we interpreted pairwise correlations using a heatmap approach in which only positive correlations were considered relevant and Spearman’s rho (r_s_) was interpreted as more likely being an important relationship via higher positive correlation coefficients. All analyses were performed using R software version 4.2.0 (R Core Team (2023). R Foundation, Vienna, Austria), using the DescTools and irrCAC packages [21].

## 3. Results

### 3.1. Demographics and Clinical Characteristics

One hundred and thirty-three patients with a compatible history of IgE-mediated banana allergic reaction were identified for eligibility assessment and recruited. The baseline characteristics of the participants are summarized in Table 1. The median current age of the patients was 36.0 years (IQR 32.0, 41.0). The median age of banana allergy onset was 33.0 (IQR 29.0, 38.0) years. A total of 96 of the 133 (72.2%) patients were female. Eighty-eight patients (66.2%) had a history of atopic-related disorders, most of which were allergic rhinitis (58.6%), chronic urticaria (9.8%), and asthma (6.8%). Eighty-three (62.9%) and forty-nine (37.1%) patients reported banana-associated first reactions, which were compatible with anaphylaxis and non-anaphylaxis, respectively. One hundred and sixteen (87.9%) patients had at least one reported anaphylactic episode related to banana ingestion. The banana cultivar and common recipes containing bananas which were commonly consumed by the study population are summarized in Figure A1. The proportions of banana cultivars consumed during first reaction were Pisang Awak (44.4%, 58/133), Cavendish (43.6%, 58/133), Leb-mue-nang (5.3%, 7/126), Lady’s fingers (4.5%, 6/133), and Silver Bluggoe (0.0%, 0/133).

Of the 98 patients reporting a history of using latex-containing products, only 8 (8.1%) reported immediate reactions, i.e., contact urticaria, angioedema, and anaphylaxis.

### 3.2. PTP and Banana-Specific IgE Results

Of the 133 patients, 130 (97.7%) showed positive PTP results to at least one banana cultivar, while 80 out of 123 patients (65.0%) showed positive banana-specific IgE using a cut-off point of >0.35 kU/L. (Table 1)

### 3.3. Clinical Reactions to Other Plant Foods

Of 133 patients, 117 (87.9%) banana-allergic patients reported a history of other cross-reactive plant food allergies. The plant foods most commonly reported for cross-reactivity were kiwi (*n* = 76; 83.5%), avocado (*n* = 64; 71.1%), grape (*n* = 51; 44.0%), durian (*n* = 48; 43.6%), and persimmon (*n* = 40; 58.8%) (Figure 2).

The proportions of clinical cross-reactivity/co-allergy to other plant foods in banana-allergic patients are summarized in Table 2. The proportion of patients with a clinical allergy to at least two plant foods was 51.8%. The proportions of non-severe and severe reactions (anaphylaxis) were 73.5% and 26.5%, respectively.

### 3.4. PTP for Possible Cross-Reactive or Co-Allergic Plant Foods

PTP was performed on kiwi and avocado, as well as green and red grapes. The results of PTP to predict clinical cross-reactivity or co-allergy to kiwi, avocado, green grape, and red grape are summarized in Figure A2. Banana-allergic patients have specific plant food allergic reactions despite negative PTP to kiwi, avocado, and grape, which were 72.7% (8/11), 52.4% (11/21), and 21.6% (8/37), respectively. Moreover, patients with positive PTP without any reaction after the ingestion of kiwi, avocado, and grape, were 15.2% (12/79), 23.2% (16/69), and 43.7% (31/71), respectively.

### 3.5. Correlation Analysis of PTP Mean Wheal Diameters

The results of Spearman’s correlation analysis for mean wheal diameters obtained from the SPT of fresh Thai banana cultivars, related fruits, plant foods, and aeroallergens are presented in Figure A3. Among all the correlations, the positive correlations between Thai banana cultivars and kiwi, avocado, green grape, red grape, and durian were the strongest (r_s_ = +0.30 to +0.79). Additionally, there were weaker positive correlations observed for soybean and peanut (r_s_ for soybean = +0.12 to +0.26, and r_s_ for peanut = +0.07 to +0.23). In particular, the Cavendish cultivar exhibited higher positive correlations with kapok (r_s_ = 0.24) and German cockroach (r_s_ = 0.20), which were not observed in the other Thai banana cultivars.

## 4. Discussion

To the best of our knowledge, this study constitutes the first examination of the clinical characteristics of banana allergy in adults using a relatively large sample size. Our study aimed to determine the proportion of clinical reactions to other plant foods in patients with banana allergy. Our findings reveal that kiwi, avocado, persimmon, grape, and durian were the plant foods most frequently associated with reported reactions. Notably, approximately 26.5% of the reported reactions to other plants were indicative of anaphylaxis. However, our findings also suggest that other plant foods may be involved in cross-reactive or co-allergic reactions.

Banana is one of the most popular fruits worldwide, especially in Asia, Latin America, and Africa, according to the Food and Agriculture Organization (FAO) of the United Nations [22]. Thailand is a relatively large-scale banana producer in Asia, with an annual increase in production capacity, mainly through domestic consumption [23]. The diagnosis of banana allergy in adult patients is also increasing worldwide [24]. Suriyamoorthy et al. [7] and Fernández-Rivas [25] reported that banana allergic syndromes are most commonly pollen-fruit syndrome, followed by latex-fruit syndrome, and lipid transfer proteins (LTP) syndrome. Oral allergy syndrome is the most common allergy syndrome, but case series reported from Thailand and some latex-fruit syndrome cases describe banana-related anaphylaxes [5]. Interestingly, in our study, we found a high rate of banana-associated anaphylaxis (87.9%) in the study population, which contradicts previous studies [5,15,26]. Typically, we consume fresh or raw fruits, which can expose sensitized individuals to both heat-labile and heat-stable allergens. Approximately 93% of patients with banana allergies experienced symptoms after consuming raw bananas, while only 15% to 59% of the patients experienced symptoms after consuming processed bananas, in which heat-labile allergens were modified.

There has been an ongoing debate concerning the necessity for patients with fruit allergies to avoid other plant foods. The potential for cross-reactivity arises due to the presence of homologous proteins shared among various plant foods. When individuals become sensitized to specific allergens, IgE antibodies can exhibit cross-reactivity with plant foods that contain cross-antigens, sharing a minimum of 70% of the primary amino acid sequence. Moreover, IgE antibodies can demonstrate cross-reactivity with multiple allergenic proteins that possess identical or similar epitopes, even in the absence of prior contact or sensitization [27].

One established mechanism through which cross-reactive plant food allergic symptoms arise is through primary sensitization to pollen food allergy syndrome (PFAS) via inhalation. In our cohort, the observed cross-reactivity patterns strongly suggest the presence of PFAS, as the predominant symptoms align with those of oral allergy symptoms and a high proportion of comorbid allergic rhinitis. Additionally, the cross-reactivity pattern exhibited by the patients closely resembled the association between the Ficus tree and various fruits. The Ficus tree (*Ficus benjamina*), also known as the weeping fig, holds the distinction of being the official tree of Bangkok, where the majority of our patients reside, and is highly prevalent in the local environment. Sensitization to Ficus tree pollen, a common indoor allergen originating from ornamental plants, can occur in both atopic and non-atopic individuals [16]. Individuals sensitized to *Ficus benjamina* have exhibited positive skin tests for various tropical and exotic fruits, including fig, papaya, banana, and pineapple, despite most of them testing negative for natural rubber latex [16]. Avocado, kiwi, and jackfruit have also been associated with Ficus-fruit syndrome [28,29]. In a previous study, individuals sensitized to *Ficus benjamina* demonstrated positive PTP for fresh fig in approximately 83% of cases, and 47% of patients experienced symptoms upon exposure [16]. Ficus-fruit syndrome is considered a distinct syndrome that is separate from natural rubber latex allergy. The primary allergens associated with Ficus-related allergies are thiol proteases rather than hevein-like proteins. Cross-reactivity has been observed among ficin (from fig), papain (from papaya), bromelain (from pineapple), and actinidin (from kiwi). These allergens share similarities and can lead to allergic reactions in individuals sensitized to any of these fruits [30]. Preliminary results from immunoblots revealed IgE binding to Ficus tree pollen extract (unpublished data). In addition, the relationship between ragweed *Ambrosia artemisiifolia* pollen and food allergies, known as the ragweed–melon–banana association, has been observed in patients with respiratory allergies to ragweed. Cross-reactivity between ragweed pollen and certain foods was first documented almost 50 years ago [31]. The ragweed–melon–banana association was named as such due to the involvement of gourd family members, such as melon, watermelon, zucchini, and cucumber, as well as banana. This association is likely attributed to sensitization to profilin, although the role of non-specific lipid transfer proteins (ns-LTPs) and glycoallergens cannot be entirely ruled out [6]. Further investigations, such as the inhibition of IgE-binding, could confirm IgE cross-reactivity and suggest the primary sensitizers.

Another relevant condition is latex fruit syndrome. However, in our study population, only 8.3% of patients with banana allergy reported immediate reactions to latex-containing products, suggesting that the latex-fruit syndrome was not the predominant syndrome. It is important to note that the primary mechanism of sensitization is not primarily driven by natural rubber latex (NRL). Previous studies have demonstrated that approximately 28% of patients with fruit allergies also exhibit sensitization to latex [17]. Allergens derived from the rubber tree *Hevea brasiliensis*, commonly known as natural rubber latex (NRL), can trigger allergies through two main routes: the inhalation of NRL aeroallergens and direct contact with materials containing NRL. The inhalation of NRL aeroallergens or contact with NRL-containing products can lead to allergic reactions in susceptible individuals [32]. The major NRL allergens that cross-react with bananas are hevein-like proteins, specifically Hev b 8 with Mus a 1, Hev b 6 and Hev b 11 with Mus a 2, Hev b 12 with Mus a 3, and Hev b 2 with Mus a 5, giving rise to what is known as a latex-fruit syndrome. Due to the high sequence homology, the cross-reactivity between Hev b 6 in latex and chitinases in fruits, particularly in banana, avocado, kiwifruit, and chestnut, often leads to severe reactions [6]. Therefore, variations exist in the percentage, severity, and types of cross-reactivity to plant foods among patients with a banana allergy, which can be attributed to the diverse potential underlying mechanisms of cross-reactivity across different regions.

Here, we uncover the clinical cross-reactivity of other plant foods in patients with banana allergies. The majority of patients with banana allergy in our study were clinically allergic to more than two plant foods, with the vast majority reporting non-severe reactions to other plant foods. Structural proteins known as profilins, such as Mus a 1 in bananas, are epitopes that act as pan-allergens and can cross-react with various fruits, including melon, watermelon, tomato, banana, pineapple, kiwi, apricot, and citrus fruits [10,33,34,35,36]. They can also cross-react with tropical and exotic fruits like persimmon [36] and mango profilin [33]. Sensitization to pollen profilins only occurs in a minority of patients, and allergic reactions are typically mild, manifesting as symptoms limited to the oropharynx, referred to as oral allergy syndrome. This occurs because profilins are not resistant to thermal processing and digestion by pepsin [37,38]. Mus a 3, which is a banana allergen, is classified as a non-specific lipid transfer protein (LTP) and is known for its stability against heat and proteolysis, leading to its high allergenicity [39]. LTPs are pan-allergens that can also cross-react with fruits from the Rosaceae family, such as peach and apricot, as well as other plant foods. Similar to profilins, LTPs are associated with more severe allergic reactions [40]. Our study revealed a substantial number of patients experienced anaphylaxis, especially after consuming grapes, durian, or persimmon.

The present study revealed a higher incidence of allergic reactions to durian in patients allergic to bananas (43.64%). A previous study reported that only 18% of patients with birch pollen allergy were allergic to durian [41]. The specific allergen in durian responsible for this cross-reactivity is still unknown [42]. The complexity of cross-reactivity arises from various factors that determine its clinical relevance, including immune factors, food-related factors, and patient-specific factors [43]. The specificity, concentration, and affinity of IgE antibodies also influence clinical relevance. Additionally, the degree of homology, stability of the allergen, and the number of cross-reactive allergens contribute to a higher risk of cross-reactivity between bananas and other plant foods. Even when susceptible individuals are exposed to cross-reactive allergens in other plant foods, the presence of augmentation factors plays a significant role in the clinical implications [43]. Patients with a new onset banana allergy should be warned of the possibility of anaphylaxis from cross-reactivity or co-allergy. Few studies have examined clinical cross-reactivity or co-allergy in tropical fruits, which are widely consumed in tropical regions [24,36,42,44,45]. The current study provides practical guidance for patients with banana allergy to consider potential cross-reactivity with other fruits. Figure 3 summarizes the cross-reactivity or co-allergy of other plant foods.

The recent recommendations from the European Academy of Allergy and Clinical Immunology (EAACI) in 2022 provide important pharmacological guidance for fruit and vegetable allergy [46]. Considering the relatively low risk of systemic reactions or severe local reactions, such as the angioedema of the lips or swelling of the oral mucosa, it is advised to have emergency medication available for self-administration orally. This emergency medication may include antihistamines and, if necessary, steroids. In the event of a systemic reaction, where symptoms extend beyond the initial site of exposure, it is recommended to have epinephrine available for self-administration through the use of an autoinjector. The prompt and effective management of potentially life-threatening allergic reactions is crucial. However, our study findings indicate a notable incidence of allergic reactions following banana ingestion (87.9%, with at least one documented anaphylactic episode in the patient’s lifetime), as well as ingestion of other plant foods (26.5%). Given these results, we strongly advocate for the consideration of self-injectable epinephrine in the comprehensive management plan for patients diagnosed with a primary banana allergy. This recommendation is driven by the significant proportion of reported reactions and the extensive range of clinical cross-reactivity and co-allergy observed, which still pose challenges in accurate prediction using existing diagnostic tools.

The present study also highlights the limitations of using PTP results to guide safe consumption or avoidance of other plant foods, as positive results may lead to unnecessary restriction. While the prick-to-prick test has been shown to improve the sensitivity of SPT in patients with plant food allergies, it is important to note that the PTP can also yield both false positive and false negative rates. Our study discovered that among patients who tested positive on the prick-to-prick test for grapes and durian, 43% and 37.5% of them, respectively, did not experience any allergic reactions. False positive results in PTP can arise due to skin irritation during the testing process [19]. Another cause of false positives is non-clinically relevant sensitization, such as cross-reactive carbohydrates (CCDs), which are highly cross-reactive between inhalant allergens and plant foods. However, CCDs are considered to have low clinical relevance [46,47]. Future studies should consider incorporating multiplex assays that include specific testing for CCDs in parallel or include a CCD blocking step before testing for protein-specific IgE recognition. On the other hand, there is the potential for false negative results in PTP. Various factors may contribute to this, including variations in the abundance of allergens within different parts of fruits and the ripening status of the fruits [48]. Moreover, the performance of PTP tests can vary depending on the diagnostician involved [49]. The interval between the last allergic reaction and the PTP testing, as well as the cross-sectional nature of the testing, further influence the likelihood of encountering false negative results in PTP. These considerations underscore the importance of careful interpretation and the potential limitations of PTP in clinical practice.

Correlations between the mean wheal diameters of Thai banana cultivars and other fruits, plant foods, and aeroallergens showed positive correlations between kiwi and avocado. There were also higher positive correlations for red grape, green grape, and durian, although these results should be interpreted with caution due to fewer available SPT results for these fruits. Correlation results from this study might suggest only shared similarities of IgE epitopes but have limited value for clinical prediction.

It is important to note that the study had some limitations. Firstly, the estimation of cross-reactivity proportion in this study was based on self-reported reactions rather than OFCs. Consequently, this methodology introduces a potential bias and may not accurately reflect the precise percentage of cross-reactivity between bananas and various fruits, as conducting OFCs for multiple fruits is impractical. In future research, conducting OFCs for other fruits would be valuable in confirming the findings of this study. However, we employed a two-step interview process to obtain a comprehensive history of IgE-mediated reactions combined with evidence of banana sensitization, thereby enhancing the validity of the diagnosis. Secondly, given the cross-sectional nature of our study, we were unable to investigate whether sensitized patients who had not consumed certain plant foods would develop clinical cross-reactivity in the future, which would require a cohort study design. Additionally, there is a potential risk of recall bias, and it would have been preferable to corroborate the validity of patient-reported data from our study population with alternative sources of information. Lastly, allergen component sensitization was not investigated in our cohort. Exploring this aspect would be a fruitful area for future research, as it could offer novel insights into cross-reactivity patterns within the Thai population. Furthermore, it would be interesting to compare these patterns with those observed in other tropical fruits among various countries in the Asia–Pacific region.

## 5. Conclusions

Patients with banana allergy demonstrated a high rate of clinical cross-reactivity or co-allergy to many tropical fruits and plant foods, with severe reactions occurring in some cases. This finding has important implications for patients with banana allergy, particularly those traveling to tropical countries where these foods are commonly consumed. Patients should be informed of the risk of clinical cross-reactivity and provided with guidance on avoidance of potentially cross-reactive foods. We encouraged the consideration of carrying self-injectable epinephrine, as there is a high proportion of anaphylaxis in a quarter of reactions that occurred from other fruits after the development of banana allergy.

## Figures and Tables

**Figure 1 foods-12-02456-f001:**
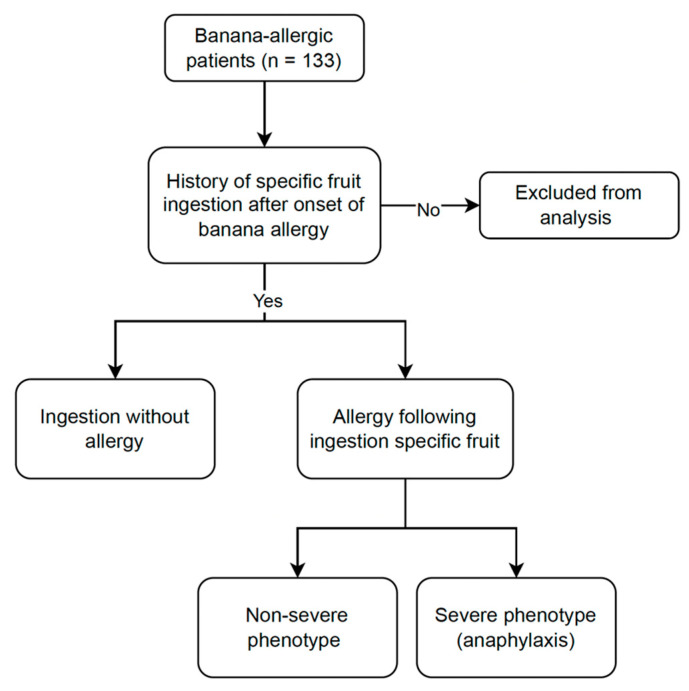
Flow chart of the patients in the study.

**Figure 2 foods-12-02456-f002:**
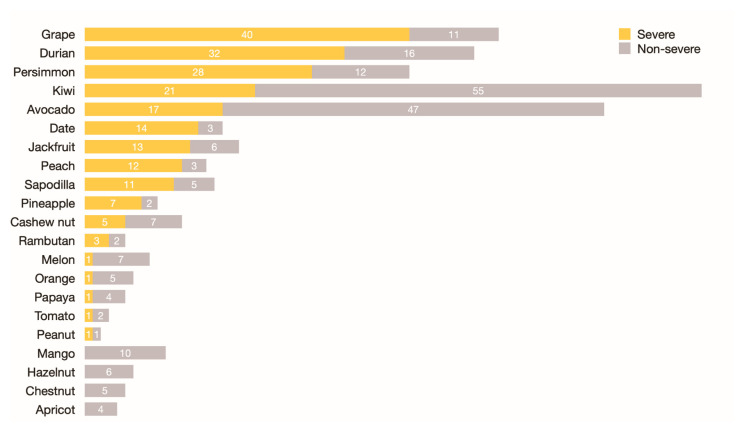
Allergic reactions and severity from cross-reactive/co-allergic plant foods in banana-allergic patients. Notes:- Clinical severity was categorized into severe reaction (anaphylaxis) or non-severe reaction (non-anaphylaxis), using the World Allergy Organization 2020 definition [20].

**Figure 3 foods-12-02456-f003:**
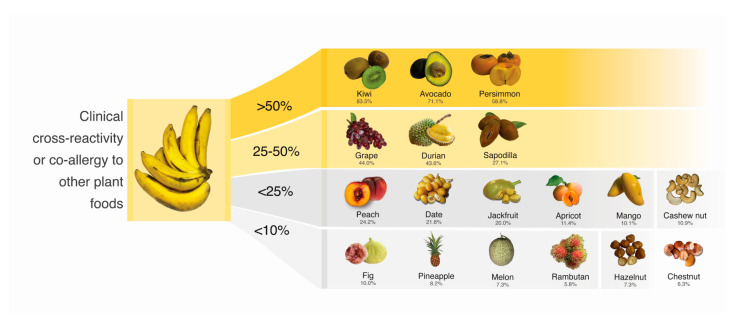
Summary of clinical cross-reactivity or co-allergy to other plant foods.

**Table 1 foods-12-02456-t001:** Patient characteristics.

Variables	All Patients (N = 133)
Female	96 (72.7)
Current age at recruitment, median (IQR), y	36.0 (32.0, 41.0)
Age of banana allergy onset, median (IQR), y	33.0 (29.0, 38.0)
Personal history of atopic-related disorders	88 (66.2)
Allergic rhinitis	78 (58.6)
Asthma	9 (6.8)
Atopic dermatitis	4 (3.0)
Chronic urticaria	13 (9.8)
Contact dermatitis	2 (1.5)
Other eczema	2 (1.5)
Reported banana-associated first reaction	
Anaphylaxis	83 (62.9)
Non-anaphylaxis	49 (37.1)
At least one anaphylactic episode related to banana ingestion	116 (87.9)
Systems involved in worst-reported reactions	
Oro-mucosal system	114 (85.7)
Generalized skin involvement (≥3 sites)	89 (66.9)
Cardiovascular system	22 (16.5)
Respiratory system	80 (60.1)
Gastrointestinal system	83 (62.4)
Positive banana prick-to-prick test ^1^	130 (97.7)
Positive banana-specific IgE ^2^	80 (65)
History of latex-related immediate reactions among latex-exposed participants ^3^	8 (8.1)

Notes: Data are presented as frequency (%) unless stated otherwise. ^1^ Positive to at least one banana cultivar.^2^ Only 123 patients had available banana-specific IgE results (ImmunoCAP, Phadia AB, Uppsala, Sweden). ^3^ Only 98 patients reported a history of using any latex-containing products. Among this group, eight patients reported immediate reactions (i.e., contact urticaria, angioedema, and anaphylaxis).

**Table 2 foods-12-02456-t002:** Proportions of other plant food cross-reactivity or co-allergy in banana-allergic patients.

Type of Plant Foods	Family	Without History of Ingestion (*n*)	Ingestion, but No Reaction (*n*)	Ingestion with Convincing Allergic Symptoms (*n*)	Proportion of Cross-Reactivity/Co-Allergy (%) ^1^
**Fruits**					
Kiwi	Actinidiaceae	42	15	76	83.5
Avocado	Lauraceae	43	26	64	71.1
Persimmon	Ebenaceae	65	28	40	58.8
Grape	Vitaceae	17	65	51	44.0
Durian	Malvaceae	23	62	48	43.6
Sapodilla	Sapotaceae	74	43	16	27.1
Peach	Rosaceae	71	47	15	24.2
Date	Arecaceae	55	61	17	21.8
Jackfruit	Moraceae	38	76	19	20.0
Apricot	Rosaceae	98	31	4	11.4
Mango	Anacardiaceae	34	89	10	10.1
Fig	Moraceae	123	9	1	10.0
Pineapple	Bromeliaceae	23	101	9	8.2
Melon	Cucurbitaceae	23	102	8	7.3
Rambutan	Sapindaceae	46	82	5	5.8
Orange	Rutaceae	6	121	6	4.7
Papaya	Caricaceae	25	103	5	4.6
Tomato	Solanaceae	14	116	3	2.5
Non-orange citrus fruits	Rutaceae	10	123	0	0.0
**Tree nuts**					
Cashew nut	Anacardiaceae	23	98	12	10.9
Hazelnut	Betulaceae	51	76	6	7.3
Chestnut	Fagaceae	51	75	5	6.3
Peanut	Fabaceae	10	121	2	1.6

^1^ The numerators were the number of patients with allergy after ingestion, and the denominators were the total number of patients with a history of ever having ingested the specific fruit.

## Data Availability

No new data were created or analyzed in this study. Data sharing does not apply to this article.

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
