# Peer review of "Not Just a Banana: The Extent of Fruit Cross-Reactivity and Reaction Severity in Adults with Banana Allergy"

_foods, 2023, doi:10.3390/foods12132456_

Round 1
Reviewer 1 Report
This paper is a descriptive cross sectional study of 133 banana allergic patients collected over a 3year period and the coross recativity with other allergens. there was high cross reactivity with kiwi, avocado and persimmon.
Rates of anaphylaxis were high and further decription of the anaphylaxis symptoms/severity would be a useful addition to the paper.
Some negative PTP subjects still had reactions - what is the mechanism for this? It would be useful to add food challenges to any future studies.
In some countries reactions to kiwi and avocado are more common, so is it possible to say what the liklihood of a kiwi positive patient reacting to banana is?
English language is good, just a few monor corrections eg line 67, line 122-123, line 194
Reviewer 2 Report
The authors wanted to look into crossreactivity between banana and other tropical fruits especially in banana allergic patients in Thailand.
Minor points: page 3 line 97: were performed
Reference 24: is this the right one for Thailand?
Methods: in the list of aeroallergens latex and ragweed is missing, they would belong to known cross-reactivity patterns
Results: correlation with aeroallergens is missing
Results 3.3 and table 2 persimmon is giving with 40 positive patient a percentage of 58.8 and for durian n=48, 43.6%? How was this percentage determined
presentation of PTP results is difficult to understand, I would like to see a better discussion why so much anaphylaxis is seen in Thailand and why the PTP is misleading
English is appropriate
Reviewer 3 Report
The authors present an interesting study on the most cross-reactive foods in patients with banana allergy.
This reviewer only has a minor question:
Could the authors offer some explanation as to why some patients are PTP negative but have a reaction after ingestions (as presented in figure A2)?
